# Integrated Taxonomy and Species Diversity of the Historical Chondrichthyan Collection of the Zoology Museum “Pietro Doderlein” at the University of Palermo (Italy)

**DOI:** 10.3390/biology14091129

**Published:** 2025-08-26

**Authors:** Maria Vittoria Iacovelli, Enrico Bellia, Martina Caruso, Ettore Zaffuto, Valentina Crobe, Federico Marrone, Stefano Mazzotti, Fausto Tinti

**Affiliations:** 1Department of Biological, Geological and Environmental Sciences, University of Bologna, 48123 Ravenna, Italy; valentina.crobe2@unibo.it (V.C.); fausto.tinti@unibo.it (F.T.); 2Museum of Zoology “P. Doderlein”, University of Palermo, 90100 Palermo, Italy; enrico.bellia@unipa.it (E.B.); martina.caruso08@community.unipa.it (M.C.); federico.marrone@unipa.it (F.M.); 3Sidercem s.r.l.-Institute for Research and Experimentation, 93100 Caltanisesetta, Italy; ettore.zaffuto@sidercem.it; 4Department of Biological, Chemical and Pharmaceutical Sciences and Technologies, University of Palermo, 90100 Palermo, Italy; 5Natural History Museum of Ferrara, 44121 Ferrara, Italy; s.mazzotti@comune.fe.it

**Keywords:** chondrichthyans, digitized catalogue, elasmobranchs, historical remains, integrative taxonomy, museum

## Abstract

Natural history collections are invaluable resources for counteracting Pauly’s “Shifting Baseline Syndrome”, the tendency to perceive a degraded environmental state as normal due to the loss of memory of past ecological conditions. These collections enable the assessment of long-term ecosystem changes and biodiversity losses. To fully leverage their potential in the digital era, it is essential to modernize legacy data systems, align collections with current museum standards, and make both data and tissue samples accessible to the global scientific community. In this study, we updated and validated the species assignments of 342 historical cartilaginous fish specimens (classes Holocephali and Elasmobranchii) housed in the Museum of Zoology “P. Doderlein” at the University of Palermo, Italy. Using an integrated 2D morphological and molecular analytical approach, we inventoried and catalogued 62 species, 42 genera, and 27 families. The resulting taxonomically validated, digitized catalogue of chondrichthyan specimens collected by Pietro Doderlein between 1863 and 1922 provides a unique temporal window into historical fish diversity in the Mediterranean. It also represents the first Italian museum fish collection aligned with current international museum standards.

## 1. Introduction

Pauly’s [1] “Shifting Baseline Syndrome” (henceforth SBS) refers to the tendency to perceive a degraded environmental state as normal, due to the loss of memory of past ecological conditions. This phenomenon arises from the temporal shift in the reference terms used to assess the condition of the natural environment. SBS can significantly hinder our understanding of long-term environmental changes and limit the effectiveness of strategies and action plans aimed at addressing current global environmental issues [2]. Furthermore, SBS contributes to the intergenerational loss of information on species abundance and diversity, with direct consequences for ecosystem conservation and restoration [3]. Incorporating historical data into environmental status assessments is therefore essential, particularly for marine populations, which often experience declines more severe than those detectable through short-term monitoring alone [3,4,5,6].

A key challenge arises in recovering historical data and samples that can be directly compared with the present-day conditions. This difficulty often stems from the degradation of primary reference environments, as well as from institutional or governmental restrictions that limit access to historical specimens, making their duplication or re-examination particularly challenging [7]. In this context of increasing unavailability of specimens and data, natural history collections, preserved in academic and civic museums worldwide, represent invaluable resources for investigating historical changes in biodiversity, species abundance, and life-history traits of the past ecosystems and taxa. These collections often provide a more comprehensive record than other data sources [5].

The huge efforts of specialists, experts, researchers, and collectors in the field—driven by the goal of documenting the biodiversity of their time—form the foundation for the preservation of specimens and associated data in museums [8]. One of the most prominent figures in 19th-century Italian marine biodiversity studies was Pietro Doderlein (1809–1895; Appendix A; [9,10,11,12,13,14]), who served as professor of Zoology and Comparative Anatomy at the University of Palermo from 1862 to 1886. In 1863, he founded the “Museo di Zoologia e Anatomia comparata” of Palermo, derived from the “Gabinetto di Storia Naturale” established in the previous year [13,14]. Over the following 31 years, Doderlein and his collaborators collected and catalogued thousands of animals and parts of them, including fossils, from Sicily and around the world, publishing important ichthyological contributions. Among these were the “Manuale Ittiologico del Mediterraneo” [15] and numerous other zoological works that demonstrate his dedication and influence in the natural sciences (Appendix A; [16,17,18,19,20,21,22,23,24,25]). Doderlein’s tireless work established the Museum of Zoology “P. Doderlein” of the University of Palermo as an internationally recognized centre for the study of historical collections. Today, the museum houses one of the best preserved ichthyological collections in the Mediterranean. Hundreds of dried fish and anatomical preparations, almost entirely collected from the northern coasts of Sicily, were gathered during the second half of the 19th century, forming the core of the anatomical [26] and the ichthyological [27] collections.

To fully leverage the potential of museum collections as reference points for assessing temporal shifts in biodiversity in the digital era, museums must address the ongoing challenge of updating and modernizing historical data systems. These systems should be aligned with current collection standards [28], ensuring all the reliable information for each specimen (e.g., collection size, conservation status, and data completeness) is accessible to the global scientific community [5]. Over the past several decades, multiple attempts have been performed by scientists, curators and cataloguers to document the anatomical [26] and ichthyological [27] collections of the Museum of Zoology “P. Doderlein”. However, each re-cataloguing effort resulted in the addition of a new label, so that many specimens now bear three or even four labels (Appendix A). This has led to inaccuracies in dating the introduction of certain specimens into the collection, thereby affecting the reliability of their legacy data. Since 1913, after being moved from their original premises in the Teatine Fathers building in Palermo’s city centre, the collection has been housed in its current location in Via Archirafi 16 in Palermo. Unfortunately, during the move, the founder’s acquisition records and manuscripts were lost partly due to a general lack of interest in the natural history collections at the time, and partly as a result of the devastation Palermo suffered during the Second World War. In 2023, a remarkable discovery was made: a personal copy of the “Prospetto Metodico” [29] containing handwritten notes by Doderlein up to 1882, along with the original register of acquisitions from 1863 to 1924. This register includes corresponding inventory numbers and complete delivery receipts detailing costs, sellers, and the provenance of each specimen.

Building on the newly retrieved archival information, this study aims to update and taxonomically validate the quali-quantitative species composition of the cartilaginous fish collection (classes Holocephali and Elasmobranchii) housed at the Museum of Zoology “P. Doderlein” of the University of Palermo. An integrated 2D morphological and molecular analytical approach was applied, and the historical chondrichthyan specimens were fully catalogued and digitized, making this the first Italian museum fish collection to be aligned with current international museum standards.

## 2. Materials and Methods

The chondrichthyan specimens of the ichthyological collection were reviewed in detail. A total of 342 complete or partial specimens are currently housed at the museum, preserved using different methods: either in liquid (alcohol or formalin) or in dry form (skull, mandible, complete skeleton, skeletal parts, or taxidermy). The documentary evidence of identification of each specimen was conducted by cross-referencing the museum’s historical catalogues with the various categories of associated historical labels (Appendix A).

### 2.1. Comparative Assessment of the Historical Records

First, a critical assessment of the historical records compiled between 1863 and 1922 was performed using Doderlein’s personal copy of the “Prospetto Metodico” [29], together with information from the anatomical [26] and ichthyological [27] collections. This review identified a total of 76 codes, corresponding to three different historical periods, each characterized by a unique numerical archiving method (Table 1). Each code, along with its categorization (Appendix A), enables the identification of the corresponding specimen.

After code identification, each specimen was examined according to several criteria. First, the presence of an inventory code was verified to determine whether the specimen could be assigned to one of the three historical periods, thereby confirming its current inclusion in the collection. For each specimen recorded in the historical register, its presence in both the “Manuale ittiologico del Mediterraneo” [15] and the “Prospetto Metodico” [29] was also assessed. This comparison relied on the species name indicated on the historical label category 2 (Appendix A), along with the measurements reported and catalogued for each specimen, allowing the integration and verification of the related information.

Specific measurements were newly collected for all specimens. The total length (TL), defined as the distance from the anterior extremity of the individual’s head to the tip of the caudal fin, and the length (L) defined as horizontal measurement of the specimen, were recorded to allow comparison with the data reported in the “Manuale ittiologico del Mediterraneo” [15]. In addition, to support the digitization and management of the collection, three spatial dimensions were recorded in accordance with the general catalogue of cultural heritage, which evaluates the spatial requirements of each specimen within the collection [30]:Height (H): the maximum vertical distance from the lowest point of the specimen (or its supporting base) to its highest point.Length (L): the horizontal distance from the leftmost to the right extremity of the specimen.Width (l): the depth, or front-to-back dimension, of the specimen’s spatial footprint.

These measurements enabled a detailed morphological and spatial characterization of each specimen, supporting its cataloguing, conservation, and integration into the collection’s digital database [30]. Sex and maturity stage were assessed for specimens preserved as taxidermized or with complete skeletal systems. Sex determination was based on the presence or absence of pelvic claspers, a distinguishing feature of male chondrichthyans. Maturity stage was estimated using literature references and the recorded body lengths [31,32,33]. Based on these assessments, specimens were classified into five categories: U (egg)—individuals still in the embryonic stage within the egg; N (foetus)—specimens not fully developed but nearing hatching or birth; J (juvenile)—individuals that are born but not yet sexually mature; SA (subadult/immature)—specimens close to sexual maturity but not fully developed; A (adult)—individuals that have reached full sexual maturity.

### 2.2. Morphological and Molecular Species Identification

Whenever possible, each specimen was identified to the species level. Morphological analysis involved comparing individuals of closely related species to detect potential misidentifications. The identification of taxidermized specimens and those with an entire internal skeleton was performed based on keys available in the literature [26,27,29,30,31]. Studied characters included the relative positioning of the dorsal fins in relation to the caudal fins, an important diagnostic trait in some genera, such as *Squatina* spp. [34,35]. Jaws were identified by analyzing tooth morphology, considering features such as crown length, cusp inclination and width, presence of secondary or multiple cusps, potential serration of the cusp, and ontogenetic variation in tooth shape between juvenile and adult individuals [34,36,37,38,39,40,41,42,43]. For osteological specimens where morphological identification was insufficient or uncertain, molecular identification was performed on drilled and powdered cartilage of specimens collected following the protocol described in [44]. All laboratory procedures adhered to strict sterility protocols to prevent contamination by exogenous DNA [45,46]. DNA extraction and pre-PCR preparation were carried out in physically separated and designated areas (pre-PCR laboratory at the Ancient DNA Laboratory of the Department of Cultural Heritage, University of Bologna, Italy), a facility dedicated solely to ancient DNA analysis [47]. Reusable tools and equipment surfaces were cleaned with bleach and ethanol or using the DNA-Exitus Plus™ cleaning kit (Applichem Inc., Omaha, NE, USA). All reagents and plasticware used during DNA extraction and pre-PCR preparation were exposed to UV radiation for 30 min before use (except for DNA polymerase, primers, and dNTPs). During ancient sample processing, laboratory personnel wore appropriate disposable protective clothing, including full suit, cap, boots, face mask, face shield, arm covers, and two pairs of gloves.

Total genomic DNA was extracted from 50 to 70 mg of cartilage powder using a chemical extraction protocol [48], carefully optimized based on previous studies [49]. Powdered tissue was obtained by drilling the most hidden part of the jaws (i.e., the Meckel’s/hyoid arch), specifically behind the joints. For taxidermied specimens, tissue samples were collected by drilling the inner part of the mouth, avoiding the preserved outer tissue. DNA was eluted in 50 µL of EB buffer (Qiagen, Hilden, Germany). To prevent contamination from amplicons, all PCR amplifications were carried out in a physically separated area dedicated exclusively to post-PCR procedures. We carried out PCR amplification by targeting a short 12S ribosomal RNA fragment (about 190–230 bp long) of the mitochondrial genome [50] using the Elas02 primers (Thermo Fisher Scientific, Waltham, MA, USA), specifically developed for elasmobranchs in eDNA samples [51,52,53,54]. PCR reactions were performed for the first time using aDNA collected from museum specimens; in 25 μL reactions, containing 3 μL of template DNA, 5 μL of 10X PCR Buffer (Invitrogen^®^, Thermo Fisher Scientific, Waltham, MA, USA), 2.5 μL of MgCl_2_ (50 mM), 1 μL of each primer (10 μM), 2 μL of dNTP mix (10 mM each) and 0.125 μL of 5U Recombinant Taq DNA Polymerase (Invitrogen^®^, Thermo Fisher Scientific, Waltham, MA, USA). Amplifications were performed using PCR 2720 Thermocyclers (Applied Biosystems™, Thermo Fisher Scientific, Waltham, MA, USA) using the following conditions: an initial DNA denaturation at 95 °C for 5 min followed by 40 cycles at 95 °C for 30 s, 60 °C for 45 s, 72 °C for 30 s, followed by a final extension of 5 min at 72 °C. Then, amplification products were checked on 2% agarose gel. Amplicons were enzymatically purified using the ExoSAP Express PCR Product Cleanup following available protocol (https://www.thermofisher.com/order/catalog/product/75001.200.UL accessed on 8 July 2025) and were sequenced in both directions by an external sequence service provider (Macrogen Europe B.V, Amsterdam, The Netherlands). Negative controls were included during both DNA extraction and PCR amplification. No amplification was observed in any of the negative controls, confirming the absence of contamination. Trace files were checked and edited with MEGA v.11 [55]. The resulting sequences were compared to published references via the GenBank BLAST algorithm (https://blast.ncbi.nlm.nih.gov/Blast.cgi accessed on 12 August 2025) and the BOLD Identification Engine [56] to evaluate their clustering within established barcode reference groups. Reference sequences showing the highest similarity (>98%) to our data were downloaded and aligned using the ClustalW algorithm (MEGA v.11) [57]. A Maximum Likelihood (ML) phylogenetic tree was subsequently reconstructed in MEGA v.11, applying the Tamura 3-parameter substitution model, which was identified as the best-fit model. Node support was evaluated through 1000 bootstrap replicates.

### 2.3. Digitization of the Chondrichthyan Collection

The digitization of the chondrichthyan specimens was carried out following the guidelines provided by the Italian Ministry for Cultural Heritage and Activities, specifically those from the Central Institute for the Cataloging and Documentation of Naturalistic Assets—Zoology [30]. The official reference document (https://catalogo.beniculturali.it/search/typeOfResources/NaturalHeritage?refineQ=pesci&startPage=24&paging=true accessed on 13 August 2025) was thoroughly reviewed to identify both mandatory and optional fields most relevant for enhancing and organizing the collection (Appendix A). In total, 13 categories and 108 structured subfields were considered for the digitization process (Appendix A).

In parallel, specimen photographs were captured using a Canon EOS 750D camera (Canon Inc., Tokyo, Japan) equipped with an 18–55 mm lens. Each specimen was photographed alongside a ruler for scale and labelled with its current cataloguing identification number. Images were processed using Adobe Photoshop 2022 (version 23.5) [58] and Adobe Lightroom Classic (version 12.0) [58] to enhance light exposure and improve the visibility of morphological details. Specifically, shadow areas were reduced, and contrast between light and shadow was increased to enhance the overall visibility of the specimen. Photographs were generally taken in a dedicated room with a white backdrop. For specimens that were large, heavy, difficult to move, or stored in inaccessible or closed display cases, on-site photography was performed. In these cases, Adobe Photoshop (version 23.5) [58] was used to extract the specimen from its original background and place it onto a white backdrop to ensure visual uniformity across the collection’s images.

Finally, a complete digitized catalogue of the chondrichthyan collection at the Museum of Zoology “P. Doderlein” was compiled. Each specimen was catalogued and documented following the format outlined below in accordance with the Eschmeyer’s Catalog of Fishes Classification, accessible at https://www.calacademy.org/scientists/projects/eschmeyers-catalog-of-fishes (accessed on 13 August 2025).

-Class (Holocephali/Elasmobranchii)-Superorder (Batoidea/Selachimorpha)-Order-Family and author-Genus and author-Species and author-Collection number-Year of acquisition-Historical number-Specimen (Digestive system/Respiratory system/Ocular system/Branchial and circulatory system/Tail/Skull/Taxidermized specimen/Jaw/Oviduct/Rostrum/Skeleton/Splanchnocranium)-Sex (♂ or ♀), maturity stage (juvenile, sub-adult, adult), length/total length (cm).

## 3. Results

### 3.1. Comparative Assessment of the Historical Records

Comparison between the specimens currently housed in the museum and those recorded in historical documents revealed discrepancies in the number of registered specimens across different periods. Of the 76 chondrichthyan codes, 40 remained consistent over time (Appendix A), while 36 showed changes (Table 2). A notable discrepancy concerns five codes (129, 151, 152, 153, and 154), which were consistently recorded between 1880 and 1922, but refer to specimens no longer present today (Table 2). In the acquisitions register from 1880 to 1890, two codes (595 and 1120) refer to specimens recorded at acquisition that are still physically present in the current collection but were missing from the 1922 inventory. Additionally, seven specimens (codes 119, 487, 564, 638, 693, 710, and 1062), although recorded at acquisition, are absent from both the 1922 and 2024 inventories (Table 2). Further inconsistencies involve 11 specimens registered between 1880 and 1890 and confirmed in the 1922 inventory but missing today. Lastly, two codes recorded in the 1922 inventory (2648 bis and 2650 bis) show a higher number of specimens in the 2024 inventory than in the earlier records from 1880 and 1922 (Table 2).

### 3.2. Morphological and Molecular Species Identification

Species identification was performed for a total of 288 specimens (Appendix A). Of these, 282 were identified through morphological analysis, based on dental morphology or external characteristics (Appendix A). A total of 58 discrepancies were found between the species names indicated on the original labels and those resulting from our identifications (Table 3 and Figure 1). However, species-level identification remained uncertain for 11 specimens (Table 3). Morphological identification at the species level was not possible for six specimens because of insufficient diagnostic features, and one specimen could not be identified because it was inaccessible (Appendix A). Among the 16 specimens subjected to molecular identification, which included the five specimens not identifiable morphologically and the 11 with uncertain morphological IDs, high-quality 216 bp 12S sequences were obtained for only eight individuals only (AN39, AN 83, AN140, AN405, AN406, P606, P616, P651; Table 3) These sequences showed very high similarity scores in BLAST searches against the NCBI GenBank database (https://www.ncbi.nlm.nih.gov/ accessed on 12 August 2025 and the BOLD Identification Engine (99.51–100%; Appendix A), supported by strong bootstrap species-specific values in the ML phylogenetic tree (61–100%; Appendix A). The ML tree was constructed using 26 reference sequences representing the closest matches identified via BLAST and BOLD searches, with *Prionace glauca* (GenBank Accession Number PQ469261) included as the outgroup. The final alignment comprised 216 positions, of which 67 were polymorphic. Among the successfully identified specimens it is noteworthy that the molecular identification of *Dipturus batis* represents the first molecular record of this species in the Mediterranean. The sequence from specimen AN39 fully matched two Atlantic reference sequences available in GenBank (accession numbers EF081271.1 and EF081277.1). Particular attention should also be given to specimens P606 and P616, originally labelled as *R. montagui* in the Doderlein’s collection. These were identified as *R. polystigma* by GenBank blasting and as *R. radula* by the BOLD system. The ML phylogenetic tree clearly clustered these specimens with the BOLD reference sequences of *R. radula* and the GenBank as *R. polystigma* (Accession Number EF100185), supporting the unambiguous identification of P606 and P616 as *R. radula*.

### 3.3. Catalogue Digitization

A total of 342 chondrichthyan specimens are currently catalogued in the “P. Doderlein” museum, representing 62 species across 27 families (Table 4). The family Rajidae is the most abundant, comprising 96 individuals belonging to 13 species. Examples of different preservation methods used for the specimens are illustrated in Appendix A.

The catalogued 342 chondrichthyan specimens, complete with validated data and information are reported in the Appendix A. Each specimen has a catalogue card available on the website https://catalogo.beniculturali.it/search/typeOfResources/NaturalHeritage (accessed on 13 August 2025; currently under scientific review).

## 4. Discussion

This study presents a taxonomically validated, digitized catalogue of the chondrichthyan collection preserved in the “P. Doderlein” Museum of Zoology of the University of Palermo, offering a unique temporal window into the historical chondrichthyan diversity of the Mediterranean from 1863 to 1922. While numerous marine taxa catalogues are reported in the literature, these typically focus on single institutions [60,61,62,63]. In Italy, national-level efforts have primarily centred on cetaceans [64]. Unlike cetological collections, which are well documented at both national and institutional levels [61,62,64], ichthyological collections housed in Italian museums have received limited attention. The creation of a well-structured, complete and taxonomically validated catalogue is essential to enhance the scientific value of Italian museum collections by making specimen data widely accessible and usable to the global research community. Although recent years have seen increased public research aimed at valorizing historical collections for the scientific community [65,66,67,68,69], a significant gap remains in accessing comprehensive data and tissue samples from other Italian museums due to the lack of comprehensive and detailed catalogues for ichthyological collections, especially those containing cartilaginous fish specimens.

The Zoological Museum “Pietro Doderlein” of the University of Palermo houses one of the largest collections of Chondrichthyes among Italian museums, comprising 342 specimens representing 62 species across 27 families. This extensive collection makes it an ideal case study for the creation of Italy’s first elasmobranch catalogue. The catalogue’s development began with a historical evaluation of the museum’s specimens, focusing especially on the original collection assembled by P. Doderlein in 1880. Since then, the collection has been managed by various curators and studied by numerous scientists. In 2023, Doderlein’s personal copy of the “Prospetto Metodico” [29] was recovered, and during the most recent collection revision in 2024, this document facilitated the accurate linking of confirmed acquisition dates and provenance to each historical specimen. When combined with data from the “Manuale ittiologico” [15], it enabled a detailed reconstruction of the history of individual specimen. Several discrepancies emerged regarding the number of specimens associated with specific inventory codes. A total of 36 out of 76 codes showed changes in the number of individuals during the years (Table 2). The earliest significant differences appeared between the 1863–1880 records and the additions up to 1922. Five codes showed minimal differences in specimen numbers (e.g., codes 118 and 153), while others (such as codes 120 and 121) exhibited slightly larger discrepancies. A more substantial variation was observed for the code 151, likely due to unregistered new acquisitions. Furthermore, seven new codes introduced by 1922 could explain discrepancies and the absence of specimens corresponding to nine of the 53 codes listed in the 1880–1890 acquisition register. These new codes were often assigned to remains with damaged, illegible, or missing labels. Additional inconsistencies were found between our revision and the historical register from 1863 to 1880. Notably, specimens associated with codes 151 (stomachs and intestines prepared as dry specimens), 152 (gills prepared as dry specimens), 153 (injected gills), 154 (hearts prepared dry) (Appendix A) were missing. These codes encompass the entire ichthyological collection, including both elasmobranchs and bony fishes, which likely contributed to classification confusion over time. Code 129 assigned to sharks and rays in alcohol, was also absent in our revision, probably because the glass container of the specimens lost the label during the years and was never replaced. Conversely, a few specimens missing from 1922 additions but recorded during initial acquisition were rediscovered during re-cataloguing, following careful cleaning and removal of paint covering original labels. Moreover, 16 specimens lacked legible or category-specific labels, preventing their definitive association with any historical acquisition. Lastly, 18 specimens preserved as taxidermized, skeleton and jaw, were missing from the current collection. These losses are likely attributable to two major causes: first, during World War II, the museum was largely unattended and insufficiently supervised, leading to theft and destruction of numerous specimens, particularly from the ichthyological and ornithological collection. Second, general neglect in the 1950s resulted in the consolidation of all collections into a single room, where careless handling by staff caused damage, loss, and further theft.

Following the reassignment of historical codes, the majority of specimens (*n* = 288) were taxonomically re-evaluated and successfully identified, while a subset (*n* = 54) remained unidentifiable due to their preservation or the lack of diagnostic features. Specimens consisting of stomachs, intestines, gills, hearts, and reproductive organs prepared as dry specimens, were excluded from the analysis due to the morphological similarity of internal organs across species, genera and families, as well as the degradation and destruction of DNA caused by the preservation methods used [70,71]. Additionally, specimens preserved in alcohol were excluded from the integrated morpho-molecular analysis due to prior formalin fixation, which degrades DNA [71] and the bleaching effects caused by subsequent alcohol preservation.

Morphological analysis of body and teeth shape, conducted on 138 specimens belonging to the order Selachimorpha, alongside genetic analysis conducted on eight specimens revealed 35 cases of species misidentifications. Notably, all specimens were correctly identified by morphological analysis, with three of these identifications further confirmed through molecular analysis. The most frequent misidentifications involved specimens originally labelled as *Prionace glauca*, which were reclassified as *Carcharhinus plumbeus.* This confusion can be attributed to P. Doderlein’s original writings [29], where *Carcharhinus plumbeus* was not considered a resident species in the Mediterranean Sea, while the morphologically similar blue shark *Prionace glauca* was regarded as resident. The close external resemblance between these two species likely led to historical misclassification. Additional misidentifications occurred among species within the genus *Centrophorus*, as well as between *Carcharias taurus* and *Odontaspis ferox* (Figure 1), where tooth morphology, a key diagnostic feature, proved essential for accurate identification [34]. Finally, integrated morpho-molecular analysis of eight specimens from the genus *Squatina* revealed frequent misidentifications among three species, underscoring their morphological similarity [34,35,39,40] and helping to clarify their historical species diversity in Sicilian waters.

The 145 batoid specimens were divided into those belonging to the family Rajidae and others, due to the highest degree of morphological similarity among species within Rajidae [26,72,73,74]. Of the 85 Rajidae specimens, 79 were specifically identified through morphological analysis, with molecular identification confirming three of these. However, two specimens originally labelled as *R. montagui*, were reclassified as *R. radula* solely by molecular analysis (Appendix A). In total, 16 specimens were found to be misidentified and have since been taxonomically re-labelled accordingly. Notably, the integrated morphological and molecular approach assigned three specimens (AN39, P653, and P660) to *Dipturus batis*. This finding provides important evidence resolving the debate on the historical presence of *D. batis* within the Mediterranean Rajidae fauna [75], with our data confirming its presence as far back as the late 19th century. To our knowledge, no molecularly validated specimens of *Dipturus batis* have been recorded during long-term Mediterranean demersal surveys, such as the Mediterranean Trawl Survey [76,77]. Over the past 30 years, specimens of *Dipturus* spp. have typically been attributed to *Dipturus nidarosiensis* and *D. oxyrinchus* based on both morphological and genetic identification [78,79,80]. Our results may have important conservation implications, suggesting a possible local extinction of the large-sized *D. batis* in the Mediterranean. Furthermore, the integrated analysis confirms the historical presence of *D. nidarosiensis* in the Mediterranean Sea, in agreement with recent commercial and scientific surveys [73,78,79,80].

Conversely, all 60 specimens belonging to the other batoid families were correctly identified based on morphology. Among these, eight specimens underwent taxonomic revision: three specimens originally labelled as *Rhinobatos rhinobatos* were reassigned to *Glaucostegus cemiculus*, and five sawfish specimens initially identified as *Pristis pristis* were reclassified as *Anoxypristis cuspidata* (1 individual), *P. zijsron* (2 individuals) and *P. pectinata* (2 individuals) [59]. However, the historical presence of sawfish in the Mediterranean remains debated, and ongoing investigations aim to clarify the geographic origin of these museum specimens [29,59].

In addition, all historical data and specimens have been catalogued according to the current systematics of Holocephali and Elasmobranchii and made publicly available online to the scientific community. Each specimen is listed in alphabetic order by its modern catalogue number, alongside the corresponding historical code when available (Appendix A).

This work ultimately provides a foundation for the historical reconstruction of the Sicilian marine biodiversity, offering a valuable resource for biodiversity studies [5] by establishing historical baselines that enable investigation of long-term ecological trends, anthropogenic impacts, while laying the groundwork for future SBS assessment studies.

### Limitations

This study faced common challenges associated with the use of historical museum specimens. Limited access to some materials (e.g., specimens preserved in closed display cases), time constraints for individual specimen identification, and the need for careful cross-verification with historical registers and manuals complicated the process. Identification of dry-preserved specimens, such as digestive tracts, was particularly difficult due to morphological similarities across families and the varied preservation methods (e.g., dry-mercury-treated, formalin-fixed), which affected both morphological and molecular analyses [70,71]. DNA extraction from historical hard tissues (e.g., the Meckel’s cartilage) required stringent laboratory protocols to prevent contamination from exogeneous DNA from fungi or bacteria [45,46]. Moreover, the preservation state and original specimen quality strongly influenced the success of downstream DNA amplification and sequencing, impacting the reliability of comparisons with reference sequences from public sequence databases such as NCBI and BOLD. Although the mitochondrial 12S fragment targeted by the ELAS02 primers is shorter than the standard 650 bp COI barcode commonly used for fish identification, its use was necessary due to the highly degraded DNA typical of historical specimens. This short 12S fragment, developed for elasmobranch detection in environmental DNA studies [51,52,53,54], is particularly suitable for fragmented and degraded DNA recovered from historical and environmental samples. We acknowledge that reference sequences for this marker remain less abundant in public databases; however, in our dataset, the 12S fragment provided sufficient polymorphic sites to resolve taxa diversity. The application of a Maximum Likelihood phylogenetic reconstruction combined with curated BOLD references helped to minimize the misidentification risk inherent in similarity-based methods alone. Despite these challenges, we believe that addressing such methodological complexities was essential, and that the results presented here are robust and well-supported by the data obtained.

## 5. Conclusions

Unlike other marine flagship taxa such as cetaceans, elasmobranchs have historically received limited attention from both the public and the scientific community. Consequently, efforts to catalogue elasmobranch’s museum collection have been minimal. This study helps fill that gap by establishing the first comprehensive, referenced, and digitized catalogue of chondrichthyan specimens across Italian museums, providing validated species identification for each specimen. Our work highlights the potential of historical collections to generate new scientific insights while enhancing the cultural and scientific value of museum holdings. Although some gaps in taxonomic identification of museum specimens remain, mainly due to the nature and condition of the specimens, this study emphasizes the critical importance of creating detailed and accurate catalogues. Such catalogues are essential for improving accessibility, fostering further research, and promoting the broader valorization of natural history collections.

## Figures and Tables

**Figure 1 biology-14-01129-f001:**
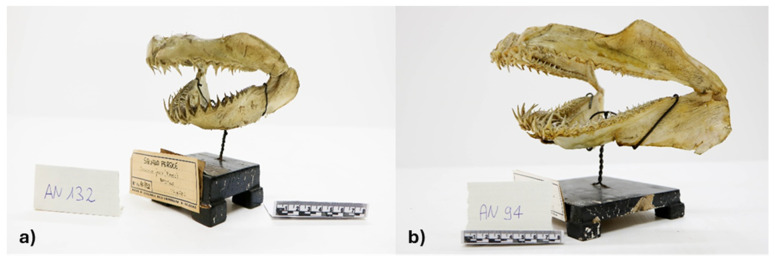
Example of mislabeled jaws. (**a**) Jaws of *Carcharias taurus* previously labelled as *Odontaspis ferox*; (**b**) Jaws of *Odontaspis ferox* previously labelled as *Carcharias taurus*.

**Table 1 biology-14-01129-t001:** Summary of the number of codes associated with chondrichthyan specimens (*n*) in the three historical periods related to the “P. Doderlein” collection register.

Register	*n*
Historical Record 1863–1880	16
Acquisitions 1880–1922	53
Additions up to 1922	7

**Table 2 biology-14-01129-t002:** Number of specimens per chondrichthyan code of the “P. Doderlein” museum collection that changed in the 1880, 1890, 1922 and 2024 inventories. – not found.

CODE	1880	1880–1890	1922	2024
118	2	–	1	1
119	18	–	18	16
120	37	–	32	29
121	53	–	46	41
129	23	–	23	–
139	36	–	36	33
143	27	–	27	25
147	45	–	45	42
148	3	–	3	5
151	169	–	185	37
152	96	–	96	1
153	30	–	30	1
154	9	–	9	1
158	6	–	3	1
119	–	1	–	–
435	–	1	1	–
487	–	1	–	–
564	–	3	–	–
589	–	1	1	–
594	–	1	1	–
595	–	1	–	1
638	–	1	–	–
693	–	1	–	–
695	–	1	1	–
710	–	2	–	–
744	–	1	1	–
765	–	1	1	–
805	–	1	1	–
807	–	1	1	–
852	–	1	1	–
1062	–	1	–	–
1115	–	1	1	–
1120	–	1	–	1
440-441	–	1	1	–
2648 bis	–	–	4	6
2650 bis	–	–	25	29

**Table 3 biology-14-01129-t003:** Species-level taxonomic re-assignment of the chondrichthyan specimens of the “P. Doderlein” museum through integrated morpho-molecular identification (* marks specimen whose morphological assignment was uncertain). AN: anatomical specimen; P: taxidermized specimen; –: failed.

A	Code	Specimen Label	Morphological Identification	Molecular Species Identification
AN	7	*Leucoraja circularis*	*Rostroraja alba*	
AN	31	*Squatina squatina*	*Squatina squatina **	–
AN	32	*Squatina squatina*	*Squatina squatina **	–
AN	33	*Squatina oculata*	*Squatina oculata **	–
AN	35	*Rostroraja alba*	*Raja brachyura*	
AN	39	*Dipturus batis*	*Dipturus batis **	*Dipturus batis*
AN	40	*Mustelus mustelus*	*Mustelus asterias*	
AN	41	*Squalus acanthias*	*Squalus blainville*	
AN	42	*Centrophorus granulosus*	*Centrophorus uyato*	
AN	51	*Raja asterias*	*Raja brachyura*	
AN	59	*Mustelus asterias*	*Mustelus mustelus*	
AN	69	*Raja asterias*	*Raja brachyura*	
AN	75	*Scyliorhinus stellaris*	*Scyliorhinus canicula*	
AN	76	*Aetomylaeus bovinus*	*Dasyatis centroura*	
AN	83	*Squatina squatina*	*Squatina aculeata **	*Squatina aculeata*
AN	84	*Squatina squatina*	*Squatina oculate **	–
AN	94	*Carcharias taurus*	*Odontaspis ferox*	
AN	103	*Raja undulata*	*Raja clavata*	
AN	105	*Prionace glauca*	*Carcharhinus plumbeus*	
AN	109	*Prionace glauca*	*Carcharhinus plumbeus*	
AN	110	*Prionace glauca*	*Echinorhinus brucus*	
AN	111	*Prionace glauca*	*Carcharhinus plumbeus*	
AN	117	*Prionace glauca*	*Carcharhinus plumbeus*	
AN	119	*Centrophorus granulosus*	*Centrophorus uyato*	
AN	122	*Centrophorus granulosus*	*Centrophorus uyato*	
AN	123	*Prionace glauca*	*Carcharhinus plumbeus*	
AN	129	*Centrophorus granulosus*	*Centrophorus uyato*	
AN	131	*Prionace glauca*	*Carcharhinus plumbeus*	
AN	132	*Odontaspis ferox*	*Carcharias taurus*	
AN	135	*Scyliorhinus canicula*	*Scyliorhinus stellaris*	
AN	140	*Squatina squatina*	*Squatina aculeata **	*Squatina aculeata*
AN	142	*Squatina oculata*	*Squatina aculeata **	–
AN	144	*Prionace glauca*	*Carcharhinus plumbeus*	
AN	182	*Pristis pristis*	*Anoxypristis cuspidata*	*Anoxypristis cuspidata* [59]
AN	183	*Pristis pristis*	*Pristis zijsron*	*Pristis zijsron* [59]
AN	184	*Pristis pristis*	*Pristis pectinata*	*Pristis pectinata* [59]
AN	185	*Pristis pristis*	*Pristis pectinata*	*Pristis pectinata* [59]
AN	186	*Pristis pristis*	*Pristis zijsron*	*Pristis zijsron* [59]
AN	405	*Squatina oculata*	*Squatina aculeata **	*Squatina aculeata*
AN	406	*Leucoraja circularis*	*Rostroraja alba **	*Rostroraja alba*
AN	1235	*Isurus oxyrinchus*	*Prionace glauca*	
AN	1236	*Isurus oxyrinchus*	*Odontaspis ferox*	
P	518	*Odontaspis ferox*	*Carcharias taurus*	
P	520	*Mustelus asterias*	*Mustelus mustelus*	
P	532	*Centrophorus granulosus*	*Centrophorus uyato*	
P	538	*Centrophorus granulosus*	*Centrophorus uyato*	
P	542	*Scyliorhinus stellaris*	*Scyliorhinus canicula*	
P	543	*Scyliorhinus stellaris*	*Scyliorhinus canicula*	
P	544	*Scyliorhinus stellaris*	*Scyliorhinus canicula*	
P	562	*Rhinobatos rhinobatos*	*Rhinobatos cemiculus*	
P	563	*Squatina oculata*	*Squatina aculeata*	
P	602	*Raja brachyura*	*Raja polystigma*	
P	604	*Raja brachyura*	*Raja polystigma*	
P	606	*Raja montagui*	–	*Raja radula*
P	612	*Raja asterias*	*Raja polystigma*	
P	616	*Raja montagui*	–	*Raja radula*
P	618	*Raja asterias*	*Raja brachyura*	
P	622	*Raja asterias*	*Raja polystigma*	
P	632	*Centrophorus granulosus*	*Centrophorus uyato*	
P	645	*Raja polystigma*	*Raja brachyura*	
P	649	*Rhinobatos rhinobatos*	*Rhinobatos cemiculus*	
P	650	*Rhinobatos rhinobatos*	*Rhinobatos cemiculus*	
P	651	*Dipturus nidarosiensis*	*Dipturus nidarosiensis **	*Dipturus nidarosiensis*
P	653	*Dipturus intermedius*	*Dipturus batis*	
P	660	*Dipturus oxyrinchus*	*Dipturus batis*	

**Table 4 biology-14-01129-t004:** Families of chondrichthyan specimens housed at the “P. Doderlein” museum. The number of specimens (*n*), the number of species (*s*) and the conservation method are reported (TAX = taxidermized specimens; SKE = skeleton; J = jaw; SKU = skull; R = rostrum; DP = dry preparations; LP = liquid preservation).

FAMILY	*n*	*s*	TAX	SKE	J	SKU	R	DP	LP
Alopiidae	7	1	2		4			1	
Carcharhinidae	19	3	4	1	13				1
Centrophoridae	10	2	4	1	4			1	
Cetorhinidae	2	1	1	1					
Chimaeridae	7	1	3	2				1	1
Dalatiidae	10	1	4	1	2			3	
Dasyatidae	23	3	6	3	8			5	1
Echinorhinidae	5	1	1		3	1			
Etmopteridae	4	1	2	1				1	
Gymnuridae	7	1	3	1				3	
Hexanchidae	11	2	4	2	4			1	
Lamnidae	16	3	2		13	1			
Mobulidae	3	1	1	1	1				
Myliobatidae	17	2	4	3	3	1		5	1
Odontaspididae	18	2	4	1	9	2		2	
Oxynotidae	3	1	2	1					
Pentanchidae	1	1						1	
Pristidae	6	3					6		
Rajidae	96	13	45	18	22			10	1
Rhinobatidae	14	2	8	2				4	
Scyliorhinidae	7	2	4	1	2				
Somniosidae	1	1	1						
Sphyrnidae	11	1	4	2	2			2	1
Squalidae	5	3	2	1	1			1	
Squatinidae	14	3	4	2	6			1	1
Torpedinidae	13	3	7	2				2	2
Triakidae	12	3	5	2	1	3		1	
Total	342	62	127	49	98	8	6	45	16

## Data Availability

The data supporting the findings of this study are available both in the Appendix A of this article and online through the official website of the Italian Ministry of Culture, specifically via the Central Institute for Cataloguing and Documentation (ICCD) platform, at: https://catalogo.beniculturali.it/search/typeOfResources/NaturalHeritage.

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
