# Peer review of "Integrated Taxonomy and Species Diversity of the Historical Chondrichthyan Collection of the Zoology Museum “Pietro Doderlein” at the University of Palermo (Italy)"

_biology, 2025, doi:10.3390/biology14091129_

Round 1

Reviewer 1 Report

Comments and Suggestions for Authors

In this paper the authors catalogue and identify the chondrichthyan collection of the natural museum of Palermo collected in the period of 1863 to 1922, undoubtedly the update of this information is invaluable for several specialists in taxonomy, ecology, evolution and biogeography among others. I kindly appreciate the effort of authors, I think that their work represent an important contribution, nevertheless, I have two major concerns:

  • After reading the title and the introduction, I started with too many expectations, but I finally ended with a bitter taste. Most of the introduction is related to the SBS syndrome, so I wrongly expect that the authors will compare their diversity results for the 1863 to 1922 temporal frame with the collection records of another temporal frame, such as 1990 to 2024, for example. Above all, in comparison to freshwater fishes, endemism in marine fishes is rare, so I would expect that the authors take a look at GBIF, download the records of all chondrichthyan species reported for the Mediterranean Sea, and later compare it with their catalog; also, they can make a rarefaction curve to evaluate the collection effort.  It is a simple proposition; surely, the authors will find a way to compare their collection records with actual records to evaluate the so-called SBS syndrome and determine whether it is real or not. If the authors decide not to perform the comparative analysis, then, they need to nuance (In Spanish we used matiz,  I am not sure if it’s the proper translation to what I am suggesting :D)  the introduction and change a little bit the title that at this moment sounds to much ambitious in comparison with the results presented.

  • I have major concerns with the molecular marker selected for the molecular identification. From my experience, I know that both the extraction and amplification of ancient DNA are a challenge, specifically for mitochondrial DNA that is not protected by histones and is easily hydrolyzed. Usually, DNA barcoding in fishes is performed with 5’ 650 bp of COI, authors amplified a very small fragment of the 12S rRNA, I think that the authors need to include in the discussion the possible caveats of work with this molecular marker, I don’t know how many polymorphic sites the authors could recovered in such a small fragment size. On the other hand, comparison with BLAST is not a way to properly identify an individual by DNA barcoding, similarity is not equivalent to homology. Yes, similarity is a common practice to state primary homologies, but it needs to be confirmed in a phylogenetic analysis. I take a quick look at Genbank; unfortunately, sequences for this molecular marker are not abundant. Despite, I suggest the authors to download at least the 10 first hits of blast for each one of the individuals with a molecular identification, cut to the same size, make a general alignment with an outgroup, build a phylogenetic tree, preferably with Beast or ML, to be able to evaluate either the BPP or the bootstrap support in each node.

Author Response

Overview

In this paper the authors catalogue and identify the chondrichthyan collection of the natural museum of Palermo collected in the period of 1863 to 1922, undoubtedly the update of this information is invaluable for several specialists in taxonomy, ecology, evolution and biogeography among others. I kindly appreciate the effort of authors, I think that their work represent an important contribution, nevertheless, I have two major concerns.

We thank the reviewer for the appreciation of our research work. We have addressed and accepted comments and suggestions that aim to considerably improve the manuscript (see below).

Detailed comments

Comment 1: After reading the title and the introduction, I started with too many expectations, but I finally ended with a bitter taste. Most of the introduction is related to the SBS syndrome, so I wrongly expect that the authors will compare their diversity results for the 1863 to 1922 temporal frame with the collection records of another temporal frame, such as 1990 to 2024, for example. Above all, in comparison to freshwater fishes, endemism in marine fishes is rare, so I would expect that the authors take a look at GBIF, download the records of all chondrichthyan species reported for the Mediterranean Sea, and later compare it with their catalog; also, they can make a rarefaction curve to evaluate the collection effort.  It is a simple proposition; surely, the authors will find a way to compare their collection records with actual records to evaluate the so-called SBS syndrome and determine whether it is real or not. If the authors decide not to perform the comparative analysis, then, they need to nuance (In Spanish we used matiz,  I am not sure if it’s the proper translation to what I am suggesting :D)  the introduction and change a little bit the title that at this moment sounds to much ambitious in comparison with the results presented.

Response 1: We thank the reviewer for this comment, and we agree with his concern. For this reason, we have reduced the focus on SBS in the Introduction by removing “progressive SBS” (line 70), which was initially only mentioned to provide the reader with an overview of the issue that may arise if the historical dimension is neglected. We also revised the Discussion adding the following sentence: “…, while laying the groundwork for future SBS assessment studies.” (line 521) to better highlight the purpose of this work.

However, we would like to highlight that the aim of this study is not to experimentally assess SBS, but rather to enhance and promote the value of historical collections as a background for future research works assessing the loss of species diversity. In particular, our goal was to document historical chondrichthyan diversity in Sicilian waters and to validate and catalogue this diversity to encourage future comparative studies between historical and modern metadata to assess SBS impacts. Anyway, as you suggested, we have compiled a comparative table (provided below) listing the species found in the Doderlein museum collection and those retrieved from the GBIF and the MEDITS datasets (the latter being an international bottom trawl survey carried out since 90s that yearly collects data on demersal fishery resources in the Mediterranean; https://data.jrc.ec.europa.eu/dataset/f25092c4-3f0f-449f-ba60-5fbfe385defc ). To ensure consistency, we limited the comparison to the Sicilian marine region, rather than the whole Mediterranean. At present, the comparison of available data presents several gaps and limitations. For example:

  • GBIF records are primarily based on human observations (especially for Sicilian waters) and not standardized surveys.
  • MEDITS data became available online only recently, and older records may still be missing from the public databases. These gaps could affect both the species occurrence, and the metadata associated to the specimens. Moreover, species identification during the surveys is performed often on board and relies solely on rough morphological traits, often without paying sufficient attention to species similarity. This can lead to frequent species misidentifications and an overestimation of some species relative to others (see [78]). For example, species such as Dipturus oxyrinchus and Dipturus batis, or Raja asterias and Raja clavata, are morphologically very similar especially at juvenile stage. Additionally, pelagic species such as Isurus oxyrinchus, Carcharodon carcharias, Carcharias taurus were never collected in demersal trawl survey but they are frequently documented as occasional catches and records.

Therefore, we would prefer not to include this comparison in the manuscript, as it does not align with the main objective of the article and, more importantly, because such an analysis would require a more detailed and rigorous approach.

Species

N of individuals

Doderlein' collection (1863-1922)

GBIF repository (1922-present)

MEDITS (1994-present)

Chimaera monstrosa

7

0

1355

Dipturus batis

4

0

2

Dipturus nidarosiensis

1

0

2

Dipturus oxyrinchus

7

1

583

Leucoraja circularis

8

0

19

Leucoraja fullonica

1

0

1

Leucoraja naevus

3

0

2

Raja asterias

9

6

352

Raja brachyura

7

1

19

Raja clavata

11

2

3556

Raja miraletus

7

0

6996

Raja polystigma

4

3

18

Raja radula

12

2

0

Rostroraja alba

17

2

60

Bathytoshia centroura

9

0

0

Dasyatis pastinaca

10

5

55

Pteroplatytrygon violacea

4

0

0

Gymnura altavela

7

0

0

Mobula mobular

3

0

0

Aetomylaeus bovinus

10

0

0

Myliobatis aquila

7

4

58

Glaucostegus cemiculus

4

0

0

Rhinobatos rhinobatos

10

0

0

Anoxypristis cuspidata

2

0

0

Pristis pectinata

2

0

0

Pristis zijsron

2

0

0

Tetronarce nobiliana

2

0

42

Torpedo marmorata

5

5

291

Torpedo torpedo

6

1

42

Hexanchus griseus

5

1

15

Heptranchias perlo

5

1

94

Echinorhinus brucus

5

0

0

Centrophorus granulosus

2

0

0

Centrophorus uyato

8

0

18

Dalatias licha

10

1

219

Etmopterus spinax

4

0

6258

Oxynotus centrina

3

0

23

Somniosus rostratus

1

0

0

Squalus blainville

3

0

6673

Squalus acanthias

1

0

54

Squatina aculeata

6

0

0

Squatina oculata

4

0

0

Squatina squatina

4

0

0

Carcharhinus brachyurus

2

0

0

Carcharhinus plumbeus

9

0

0

Prionace glauca

8

10

0

Galeus melastomus

1

1

0

Scyliorhinus canicula

5

3

11164

Scyliorhinus stellaris

2

3

16

Sphyrna zygaena

11

0

0

Galeorhinus galeus

5

1

2

Mustelus asterias

4

0

30

Mustelus mustelus

4

1

381

Alopias vulpinus

7

0

0

Cetorhinus maximus

2

0

0

Carcharodon carcharias

4

0

0

Isurus oxyrinchus

11

0

0

Lamna nasus

1

0

0

Carcharias taurus

7

0

0

Odontaspis ferox

11

0

0

[78] Cariani, A.; Messinetti, S.; Ferrari, A.; Arculeo, M.; Bonello, J.J.; Bonnici, L.; et al. Improving the Conservation of Mediterranean Chondrichthyans: The ELASMOMED DNA Barcode Reference Library. PLoS ONE 2017, 12(1), e0170244.

Comment 2: I have major concerns with the molecular marker selected for the molecular identification. From my experience, I know that both the extraction and amplification of ancient DNA are a challenge, specifically for mitochondrial DNA that is not protected by histones and is easily hydrolyzed. Usually, DNA barcoding in fishes is performed with 5’ 650 bp of COI, authors amplified a very small fragment of the 12S rRNA, I think that the authors need to include in the discussion the possible caveats of work with this molecular marker, I don’t know how many polymorphic sites the authors could recovered in such a small fragment size. On the other hand, comparison with BLAST is not a way to properly identify an individual by DNA barcoding, similarity is not equivalent to homology. Yes, similarity is a common practice to state primary homologies, but it needs to be confirmed in a phylogenetic analysis. I take a quick look at Genbank; unfortunately, sequences for this molecular marker are not abundant. Despite, I suggest the authors to download at least the 10 first hits of blast for each one of the individuals with a molecular identification, cut to the same size, make a general alignment with an outgroup, build a phylogenetic tree, preferably with Beast or ML, to be able to evaluate either the BPP or the bootstrap support in each node.

Response 2: We sincerely thank the reviewer for raising two important points regarding 1) the effectiveness of the used molecular marker and 2) the accuracy of the identification method.

  • In this study we used the ELAS02 primer pair developed by Taberlet et al. (2018), which targets a short fragment of the mitochondrial 12S rRNA gene specifically designed for elasmobranch species’ detection in environmental DNA studies and that has been successfully used to monitor contemporary species diversity in several field works [51-54]. This marker (that displays ca. 30% of polymorphic sites in our elasmobranch dataset) is increasingly applied to elasmobranchs in eDNA and its short size makes it particularly suitable for degraded and fragmented DNA such as that obtained from historical and environmental samples. We are fully aware that the standard DNA barcoding marker for fishes is the universal barcode ca. 650 bp-COI fragment; however, given the highly degraded nature of ancient DNA extracted from historical museum specimens, amplification of such long fragments is generally not feasible.

Nevertheless, we have acknowledged the limited power of resolution of this short 12S marker in the newly inserted “Limitations” paragraph of the Discussion (requested by the Reviewer #2), as follows (lines 532-551)”: Moreover, the preservation state and original specimen quality strongly influenced the success of downstream DNA amplification and sequencing, impacting the reliability of comparisons with reference sequences from public sequence databases such as NCBI and BOLD. Although the mitochondrial 12S fragment targeted by the ELAS02 primers is shorter than the standard 650 bp COI barcode commonly used for fish identification, its use was necessary due to the highly degraded DNA typical of historical specimens. This short 12S fragment, developed for elasmobranch detection in environmental DNA studies [51-54], is particularly suitable for fragmented and degraded DNA recovered from historical and environmental samples. We acknowledge that reference sequences for this marker remain less abundant in public databases; however, in our dataset, the 12S fragment provided sufficient polymorphic sites to resolve taxa diversity. The application of a Maximum Likelihood phylogenetic reconstruction combined with curated BOLD references helped to minimize the misidentification risk inherent in similarity-based methods alone”.

  • We are deeply grateful to the reviewer, as the suggestion to improve the identification method by incorporating ML phylogenetic analysis allowed us to correctly identify two specimens (P606 and P616) in the collection. This also enabled us to detect their molecular misclassification, which had resulted from relying on a single reference platform. We carried out the requested ML analysis by retrieving all available reference sequences from the NCBI and BOLD databases. Thus, we updated the Materials and Methods section, adding the following sentence (lines 230-238): “The resulting sequences were compared to published references via the GenBank BLAST algorithm (https://blast.ncbi.nlm.nih.gov/Blast.cgi) and the BOLD Identification Engine [56] to evaluate their clustering within established barcode reference groups. Reference sequences showing the highest similarity (>98%) to our data were downloaded and aligned using the ClustalW algorithm [57]. A Maximum Likelihood (ML) phylogenetic tree was subsequently reconstructed in MEGA v.11, applying the Tamura 3-parameter substitution model, which was identified as the best-fit model. Node support was evaluated through 1,000 bootstrap replicates.”

We then incorporated and discussed the results of this analysis (Figure S2) in the main text as follows (lines 326-332): “ These sequences showed very high similarity scores in BLAST searches against the NCBI GenBank database and the BOLD Identification Engine (99.51–100%; Table S5), supported by strong bootstrap species-specific values in the ML phylogenetic tree (61–100%; Figure S2). The ML tree was constructed using 26 reference sequences representing the closest matches identified via BLAST and BOLD searches, with Prionace glauca (GenBank Accession Number PQ469261) included as the outgroup. The final alignment comprised 216 positions, of which 67 were polymorphic.”.

Additionally, we added the following sentence to highlight a case of discrepancy in the species identification using the GenBank and BOLD databases (lines 336-342): “Particular attention should also be given to specimens P606 and P616, originally labelled as R. montagui in the Doderlein’s collection. These were identified as R. polystigma by GenBank blasting and as R. radula by the BOLD system. The ML phylogenetic tree clearly clustered these specimens with the BOLD reference sequences of R. radula and the GenBank as R. polystigma (Accession Number EF100185), supporting the unambiguous identification of P606 and P616 as R. radula.”

Accordingly, we added the ML phylogenetic tree as a new figure in the Supplementary Material (Figure S2), and we updated data on P606 and P616 in the table 3, table S4 and table S5.

[51] Taberlet, P.; Bonin, A.; Zinger, L.; Coissac, E. Sampling. In Environmental DNA: For Biodiversity Research and Monitoring; Taberlet, P., Bonin, A., Zinger, L., Coissac, E., Eds.; Oxford University Press: Oxford, UK, 2018; pp. 28–34

[52] Liu, Z.; Collins, R. A.; Baillie, C.; Rainbird, S.; Brittain, R.; Griffiths, A. M.; Sims, D. W.; Mariani, S.; Genner, M. J. Environmental DNA Captures Elasmobranch Diversity in a Temperate Marine Ecosystem. Environ. DNA 2022, 4(5), 1024–1038.

[53] Albonetti, L.; Maiello, G.; Cariani, A.; Carpentieri, P.; Ferrari, A.; Sbrana, A.; Shum, P.; Talarico, L.; Russo, T.; Mariani, S. DNA Metabarcoding of Trawling Bycatch Reveals Diversity and Distribution Patterns of Sharks and Rays in the Central Tyrrhenian Sea. ICES J. Mar. Sci. 2023, 80, 664–674.

[54] Maiello, G.; Bellodi, A.; Cariani, A.; Carpentieri, P.; Carugati, L.; Cicala, D.; Ferrari, A.; Follesa, C.; Ligas, A.; Sartor, P.; Sbrana, A.; Shum, P.; Stefani, M.; Talarico, L.; Mariani, S.; Russo, T. Fishing in the Gene-Pool: Implementing Trawl- Associated eDNA Metaprobe for Large Scale Monitoring of Fish Assemblages. Rev. Fish Biol. Fish. 2024, 34, 1293–1307.

Reviewer 2 Report

Comments and Suggestions for Authors

The study makes a valuable contribution to zoological research, particularly in the field of systematics, which often relies heavily on museum specimens. The manuscript is generally well-written and presents its findings clearly. However, I have a minor suggestion that the authors should consider including a separate "Limitations" section. In my view, this addition would strengthen the manuscript by acknowledging the common challenges researchers may face when re-examining voucher specimens related to mislabeling, taxonomic revisions, and varying preservation methods. 

Author Response

The study makes a valuable contribution to zoological research, particularly in the field of systematics, which often relies heavily on museum specimens. The manuscript is generally well-written and presents its findings clearly. However, I have a minor suggestion that the authors should consider including a separate "Limitations" section. In my view, this addition would strengthen the manuscript by acknowledging the common challenges researchers may face when re-examining voucher specimens related to mislabeling, taxonomic revisions, and varying preservation methods. 

Response:  We thank the reviewer for the minor but really important suggestion to improve our manuscript. We have taken the suggestion into account as described below by adding the “Limitations” paragraph in the Discussion (4.1 Limitations, lines 522-553) as follows:

“4.1 Limitations

This study faced common challenges associated with the use of historical museum specimens. Limited access to some materials (e.g. specimens preserved in closed display cases), time constraints for individual specimen identification, and the need for careful cross-verification with historical registers and manuals complicated the process. Identification of dry-preserved specimens, such as digestive tracts, was particularly difficult due to morphological similarities across families and the varied preservation methods (e.g., dry- mercury-treated, formalin-fixed), which affected both morphological and molecular analyses [69-70]. DNA extraction from historical hard tissues (e.g., the Meckel’s cartilage) required stringent laboratory protocols to prevent contamination from exogeneous DNA from fungi or bacteria [45-46]. Moreover, the preservation state and original specimen quality strongly influenced the success of downstream DNA amplification and sequencing, impacting the reliability of comparisons with reference sequences from public sequence databases such as NCBI and BOLD. Although the mitochondrial 12S fragment targeted by the ELAS02 primers is shorter than the standard 650 bp COI barcode commonly used for fish identification, its use was necessary due to the highly degraded DNA typical of historical specimens. This short 12S fragment, developed for elasmobranch detection in environmental DNA studies [51-54], is particularly suitable for fragmented and degraded DNA recovered from historical and environmental samples. We acknowledge that reference sequences for this marker remain less abundant in public databases; however, in our dataset, the 12S fragment provided sufficient polymorphic sites to resolve taxa diversity. The application of a Maximum Likelihood phylogenetic reconstruction combined with curated BOLD references helped to minimize the misidentification risk inherent in similarity-based methods alone. Despite these challenges, we believe that addressing such methodological complexities was essential, and that the results presented here are robust and well-supported by the data obtained.”

Reviewer 3 Report

Comments and Suggestions for Authors

The study "Integrated taxonomy and species diversity of the historical chondrichthyan collection of the Zoology Museum “Pietro Doderlein” at the University of Palermo (Italy)" by Maria Vittoria Iacovelli and colleagues addresses the important task of revising and digitising the historical collection, which is invaluable for reconstructing the basic levels of biodiversity and making data available to colleagues in the systematics community.
The authors studied 342 specimens of cartilaginous fish - 62 species from 27 families. They compared historical records, performed morphologicalverification and validated some specimens using DNA. The latter was done using ancient DNA methods.
The work makes a strong positive impression (especially the surprising discovery of a new species for the location) and undoubtedly deserves publication in the journal Biology after addressing some comments.

Comments:

1. The abstract states that 16 morphologically unidentified specimens were identified genetically, while the results and Table S5 indicate that usable 12S sequences were obtained for only 8 of the 16 specimens tested. Please clarify.
2. The methods state that molecular identification was performed on cartilage powder, but do not specify from which anatomical structure (vertebra, chondrocranium, Meckel's/hyoid arch, fin support, etc.) and where exactly the drilling was performed. This is critically important for working with ancient/museum DNA due to the high variability in yield and fragmentation depending on the tissue type. Please specify for each sample — this will be very useful information for colleagues, along with the DNA concentration provided.
3. Strict anti-contamination procedures and zone separation are described, but it is not stated whether blank extractions and PCR negatives were performed. Please clearly describe the controls, if any.
4. The text compares GenBank hits and even provides match numbers for Dipturus batis, but there is no information about the deposit of your own sequences. Please upload the 12S sequences to GenBank and provide the numbers in Table S5.
5. In the methods, TL is defined only for whole specimens, but the catalogue also includes "Total length" for skeletal parts, which can be interpreted as the biological TL of the specimen.

Comments on the Quality of English Language

Possible typos and inconsistencies:

1. "As a results" - replace with "As a result". Line 103.
2. "in contrast for those specimens of huge sizes or for those with difficult-to-access places were collected on-site" - the meaning is unclear. If possible, please rephrase. Lines 240-241.
3. In the Supplementary, I found "ovverCarcharodon". Possibly a typo.

Author Response

Overview

The study "Integrated taxonomy and species diversity of the historical chondrichthyan collection of the Zoology Museum “Pietro Doderlein” at the University of Palermo (Italy)" by Maria Vittoria Iacovelli and colleagues addresses the important task of revising and digitising the historical collection, which is invaluable for reconstructing the basic levels of biodiversity and making data available to colleagues in the systematics community. The authors studied 342 specimens of cartilaginous fish - 62 species from 27 families. They compared historical records, performed morphological verification and validated some specimens using DNA. The latter was done using ancient DNA methods. The work makes a strong positive impression (especially the surprising discovery of a new species for the location) and undoubtedly deserves publication in the journal Biology after addressing some comments.

We greatly appreciated the Reviewer’s comments and suggestions. We addressed them in the manuscript as follows.

  1. The abstract states that 16 morphologically unidentified specimens were identified genetically, while the results and Table S5 indicate that usable 12S sequences were obtained for only 8 of the 16 specimens tested. Please clarify.

Response 1: We modified the abstract coherently with the comment at lines 45-46 “Sixteen specimens that could not morphologically assigned were analyzed by DNA barcoding, resulting in eight additional species-level identifications.”

  1. The methods state that molecular identification was performed on cartilage powder, but do not specify from which anatomical structure (vertebra, chondrocranium, Meckel's/hyoid arch, fin support, etc.) and where exactly the drilling was performed. This is critically important for working with ancient/museum DNA due to the high variability in yield and fragmentation depending on the tissue type. Please specify for each sample — this will be very useful information for colleagues, along with the DNA concentration provided.

Response 2:  We detailed in the text the anatomical structures used for sampling at lines 208-211 “Powdered tissue was obtained by drilling the most hidden part of the jaws (i.e. the Meckel's/hyoid arch), specifically behind the joints. For taxidermied specimens, tissues samples were collected by drilling the inner part of the mouth, avoiding the preserved outer tissue.”

  1. Strict anti-contamination procedures and zone separation are described, but it is not stated whether blank extractions and PCR negatives were performed. Please clearly describe the controls, if any.

Response 3: We fully agree with the criticism to add the description of this routinely but mandatory experimental requirements. Thus, we changed the Materials and Methods section at lines 228-230 as follows “Negative controls were included during both DNA extraction and PCR amplification. No amplification was observed in any of the negative controls, confirming the absence of contamination.”

  1. The text compares GenBank hits and even provides match numbers for Dipturus batis, but there is no information about the deposit of your own sequences. Please upload the 12S sequences to GenBank and provide the numbers in Table S5.

Response 4: At this stage, we have already uploaded the 12S rDNA sequences in the GenBank but we’ll release their Accession Numbers only upon acceptance of the manuscript. We’ll add Accession Numbers for all sequences generated in Table S5 in the proof version before publication.

  1. In the methods, TL is defined only for whole specimens, but the catalogue also includes "Total length" for skeletal parts, which can be interpreted as the biological TL of the specimen.

Response 5: We truly appreciate this effective criticism. We did not consider the possibility of a confounding factor, so to clarify the text, we have added the following sentence at line 159 “...and the length (L) defined as horizontal measurement of the specimen.” Moreover, we revised the Appendix replacing the Total Length with Length when where appropriate.

Possible typos and inconsistencies:

  1. "As a results" - replace with "As a result". Line 103.

Response 1: Changed.

  1. "in contrast for those specimens of huge sizes or for those with difficult-to-access places were collected on-site" - the meaning is unclear. If possible, please rephrase. Lines 240-241.

Response 2: We rephrased the sentence as follows at lines 265-266 “For specimens that were large, heavy, difficult to move, or stored in inaccessible or closed display cases, on-site photography was performed.”.

  1. In the Supplementary, I found "ovverCarcharodon". Possibly a typo.

Response 3: Corrected as “ovvero Carcharodon”.

Round 2

Reviewer 1 Report

Comments and Suggestions for Authors

I kindly appreciate the effort and time invested by the authors to properly solve my major concerns. I have only have one minor comment, I know that the supplementary material is accesory, but the Figure S2 has a very low quality resolution, if the authors could improve this figure quality be great.

Felicidades